# Phytochemical Study and Antiglioblastoma Activity Assessment of *Plectranthus hadiensis* (Forssk.) Schweinf. ex Sprenger var. hadiensis Stems

**DOI:** 10.3390/molecules27123813

**Published:** 2022-06-14

**Authors:** Eva María Domínguez-Martín, Mariana Magalhães, Ana María Díaz-Lanza, Mário P. Marques, Salvatore Princiotto, Ana M. Gómez, Thomas Efferth, Célia Cabral, Patricia Rijo

**Affiliations:** 1Center for Research in Biosciences & Health Technologies (CBIOS), Universidade Lusófona de Humanidades e Tecnologias, Campo Grande 376, 1749-024 Lisbon, Portugal; evam.dominguez@uah.es (E.M.D.-M.); salvatore.princiotto@unimi.it (S.P.); 2New Antitumor Compounds—Toxic Action on Leukemia Cells Research Group, Pharmacology Area (Pharmacognosy Laboratory), Department of Biomedical Sciences, Faculty of Pharmacy, University of Alcalá de Henares, Ctra. A2, Km 33.100—Campus Universitario, Alcalá de Henares, 28805 Madrid, Spain; ana.diaz@uah.es; 3PhD Programme in Experimental Biology and Biomedicine, Institute for Interdisciplinary Research (IIIUC), University of Coimbra, Casa Costa Alemão, 3030-789 Coimbra, Portugal; mmagalhaes@cnc.uc.pt; 4CNC—Center for Neuroscience and Cell Biology, University of Coimbra, 3004-516 Coimbra, Portugal; 5Faculty of Medicine, Clinic Academic Center of Coimbra (CACC), Coimbra Institute for Clinical and Biomedical Research (iCBR), University of Coimbra, 3000-548 Coimbra, Portugal; uc2014225877@student.uc.pt; 6Center for Innovative Biomedicine and Biotechnology (CIBB), University of Coimbra, 3000-548 Coimbra, Portugal; 7Instituto de Química Orgánica, IQOG-CSIC, Juan de la Cierva 3, 28006 Madrid, Spain; ana.gomez@csic.es; 8Department of Pharmaceutical Biology, Institute of Pharmaceutical and Biomedical Sciences, Johannes Gutenberg University, Staudinger Weg 5, 55128 Mainz, Germany; efferth@uni-mainz.de; 9Centre for Functional Ecology, Department of Life Sciences, University of Coimbra, Calçada Martim de Freitas, 3000-456 Coimbra, Portugal; 10Instituto de Investigação do Medicamento (iMed.ULisboa), Faculty of Pharmacy, University of Lisbon, 1649-003 Lisbon, Portugal

**Keywords:** cancer, diterpenes, glioblastoma, Lamiaceae, phytochemical, *Plectranthus*, royleanones

## Abstract

Glioblastoma (GB) is the most malignant form of primary astrocytoma, accounting for more than 60% of all brain tumors in adults. Nowadays, due to the development of multidrug resistance causing relapses to the current treatments and the development of severe side effects resulting in reduced survival rates, new therapeutic approaches are needed. The genus *Plectranthus* belongs to the Lamiaceae family and is known to be rich in abietane-type diterpenes, which possess antitumor activity. Specifically, *P. hadiensis* (Forssk.) Schweinf. *ex* Sprenger has been documented for the use against brain tumors. Therefore, the aim of this work was to perform the bioguided isolation of compounds from the acetonic extract of *P. hadiensis* stems and to investigate the in vitro antiglioblastoma activity of the extract and its isolated constituents. After extraction, six fractions were obtained from the acetonic extract of *P. hadiensis* stems. In a preliminary biological screening, the fractions V and III showed the highest antioxidant and antimicrobial activities. None of the fractions were toxic in the *Artemia salina* assay. We obtained different abietane-type diterpenes such as 7α-acetoxy-6β-hydroxyroyleanone (Roy) and 6β,7β-dihydroxyroyleanone (DiRoy), which was also in agreement with the HPLC-DAD profile of the extract. Furthermore, the antiproliferative activity was assessed in a glioma tumor cell line panel by the Alamar blue assay. After 48 h treatment, Roy exerted strong antiproliferative/cytotoxic effects against tumor cells with low IC_50_ values among the different cell lines. Finally, we synthesized a new fluorescence derivative in this study to evaluate the biodistribution of Roy. The uptake of BODIPY-7α-acetoxy-6β-hydroxyroyleanone by GB cells was associated with increased intracellular fluorescence, supporting the antiproliferative effects of Roy. In conclusion, Roy is a promising natural compound that may serve as a lead compound for further derivatization to develop future therapeutic strategies against GB.

## 1. Introduction

Cancer is one of the main causes of morbidity and mortality worldwide. The World Health Organization (WHO) reported approximately 14.1 million new cases in 2012 and 8.2 million deaths. These figures are expected to increase due to population growth and aging, with an estimate of more than 20 million new cancer cases diagnosed per year and 13 million deaths by 2030 [1,2].

Among the different cancers, brain and central nervous system (CNS) tumors, including glioblastoma (GB), are the 12th most common tumor type according to the WHO report in 2020. It is the most frequent and malignant tumor of the CNS affecting mainly adults at the age of 40–60 [3,4].

Current treatments available for GB comprise surgical resection followed by concomitant chemoradiotherapy plus chemotherapy (temozolomide, TMZ) and the use of biological drugs, such as the therapeutic antibody bevacizumab. However, these treatments are associated with side effects and multidrug resistance (MDR), eventually yielding relapses. In fact, GB has a median survival rate of 15–18 months after diagnosis and a 5% chance of survival in the 5 years postdiagnosis. Therefore, new therapeutic approaches are needed to overcome these problems [3,5,6].

Medicinal plants are a rich source of bioactive natural products with a high structural diversity and pleiotropic effects. Therefore, they represent a potential source of anticancer lead compounds. The screening, isolation, and identification of secondary metabolites from medicinal plants have served as scaffolds for the development of derivatives with optimized potency [7]. Compared to synthetic compound libraries, naturally occurring molecules typically have higher molecular masses, a large number of sp^3^ carbons and oxygen atoms, but fewer nitrogen and halogen atoms, higher numbers of H- bond acceptors and donors, lower partition coefficients, and a greater molecular rigidity. All these characteristics make natural products valuable in drug discovery for their ability to tackle protein–protein interactions; moreover, they represent the major source of oral drugs beyond Lipinski’s rule of five [8].

Some of the most promising plant-derived anticancer compounds include diterpenes, such as paclitaxel and docetaxel [7]. Due to the importance of terpenes as anticancer drugs, the study of botanical families rich in these compounds is a promising approach to find new bioactive compounds against GB.

The genus *Plectranthus* belongs to the Lamiaceae family and consists of 300 species distributed in Africa, Asia, and Australia. The most frequently cited traditional use of *Plectranthus* species is related to their medicinal properties [9].

The first *Plectranthus* species to be described was *P. hadiensis* (Forssk.) Schweinf. *ex* Sprenger in 1775. However, it is botanically still poorly defined, resulting in identification problems until today [10,11,12]. In order to resolve these issues, some investigations have attempted to differentiate *P. hadiensis* varieties and its counterfeits by using molecular biology techniques [13,14].

In fact, the *World Flora Online* database suggests more than 30 possible names referring to this species, e.g., *P. ramosior* (Benth.) van Jaarsv., *P. cyaneus* Gürke, *Ocimum hadiense* Forssk. or *Coleus zeylanicus* (Benth.) L.H.Cramer which is considered as a variety of *P. madagascariensis* (*P. madagascariensis* var. *ramosior* Benth.). *P. hadiensis* itself also has two varieties, i.e., var. *hadiensis* and var. *tomentosus* [12,15]. A recent text-mining study that applied the novel degrees of publication (DoP) method identified *P. hadiensis* as an understudied species [16]. It was, therefore, claimed as being interesting for future laboratory studies, especially for pharmacological bioassays, isolation procedures, and drug discovery strategies.

Concerning its phytochemical analysis and biological activity, polyphenols, alkaloids, tannins, cardiac glycosides, saponins, and terpenoids have been identified in different parts, such as roots, stems, leaves, and whole extracts [17,18,19,20,21,22,23,24,25,26,27,28,29,30,31,32]. Although some of these compounds have been isolated, the phytochemistry of this genus is far from being fully known [31].

This plant is highly aromatic and possesses a distinct and specific aroma. Its essential oil is useful in ethnopharmacology practices [33] to treat various infectious diseases caused by bacteria (e.g., *Mycobacterium tuberculosis*) and fungi [19,24,28,30,34,35,36].

Other studies reported on its anti-inflammatory activity (inhibiting cyclooxygenases 1 and 2 (COX-1, COX-2) and lipoxygenases 2 and 15 (LOX-2, LOX-15) [17,31,36]. It also exerts antioxidant [20,21,22,24,32,36], larvicidal [18], insecticidal [23], and antiplasmodial activities [25,27], and is used to treat sore throat, fever, and nauseas [37].

Among all phytochemical classes, diterpenes are the most prominent biologically active group of secondary metabolites present in this genus, whose presence is considered as a chemotaxonomic marker [38]. In particular, abietane diterpenes have attracted attention regarding their biological activities, namely because of their antiproliferative, antitumoral, and/or cytotoxic potential [39,40].

To date, most phytochemical studies on *Plectranthus* species have focused on the isolation of diterpenoids. Although some of these studies reported on the biological activity of isolated diterpenoids, very few applied activity-guided fractionation approaches to isolate the compounds associated with a specific ethnobotanical use [9]. Some abietane diterpenoids have been isolated from *P. hadiensis* (Figure 1). Among them, 7α-acetoxy-6β-hydroxyroyleanone (Roy, **1**), its isomer 7β-acetoxy-6β-hydroxyroyleanone (**2**) and 6β,7β-dihydroxyroyleanone (DiRoy, **3**) have been previously characterized from the whole plant ethanolic extract [41]. 7-Formyloxy-6ß,12-dihydroxy-abieta-8,12-diene-11,14-dione (**4**) and 7α-acetoxy-6ß,12-dihydroxy-abieta-8,12-diene-11,14-dione (**5**) were isolated from the dichloromethane extract of air-dried leaves [25]. 11-Hydroxysugiol (**6**) (12-O-demethylcryptojaponol, 11,12-dihydroxyabieta-8,11,13-trien-7-one, (4a-S,10aS)-2,3,4,4a,10,10a-hexahydro-5,6-dihydroxy-1,1,4a-trimethyl-7-(1-methylethyl) phenanthren-9(1H)-one), 11,20-dihydroxysugiol (**7**) (11,12,20-trihydroxyabieta-8,11,13-trien-7-one, (4aR,10aS)-2,3,4,4a,10,10a-hexahydro-5,6-dihydroxy-4a-(hydroxymethyl)-1,1-dimethyl-7-(1-methylethyl)phenanthren-9(1H)-one), carnosolon (**8**) (6,20-epoxy-6,11,12-trihydroxyabieta-8,11,13-trien-7-one, (4aR,10S,10aS)-1,2,3,4,10,10a-hexahydro-5,6,10-trihydroxy-1,1-dimethyl-7-(1-methylethyl)-9H-10,4a(epoxymethano)phenanthren-9-one) and 1,11-epoxy-6,12-dihydroxy-20-norabieta-1(10),5,8,11,13-pentaen-7-one (**9**) were found in the diethyl ether extract of the aerial parts [42]. Coleone P (**10**) and callistric acid (**11**) (4-epidehydroabietic acid) appeared in the hexane whole plant extract [31].

Concerning its antitumoral properties, this plant has been described to be useful for the treatment of abdominal, brain, breast, intestinal, prostate, skin, throat, and uterine cancers as well as of leukemia [37]. Methanolic and ethanolic *P. hadiensis* shoot extracts revealed cytotoxicity towards brine shrimp [20,22], and stem extracts were tested against HeLa cells [26]. The terpenoid fraction (monoterpenes and sesquiterpenes) isolated from the shoot had antiproliferative activity against human colon cancer cells (HCT-15) but was only minimally cytotoxic towards noncancerous normal cells. The upregulation of caspase-3 activity and proapoptotic BAX and the downregulation of antiapoptotic BCL-2 and COX-2 indicated apoptosis induction by the mitochondria-dependent pathway [17].

Regarding the antitumoral effects of different abietane diterpenes present in *Plectranthus* species, recent results showed that abietane diterpenoids DiRoy (**3**) and Roy (**1**) induced apoptosis in glioma cell lines. These compounds exhibited antitumoral effects in primary H7PX glioma cells by inducing DNA double-strand breaks, G2/M cell cycle arrest, and apoptosis as indicated by increased levels of phosphorylated γ-H2AX, decreased mitochondrial membrane potential, and altered expression of pro- and antiapoptotic genes (*BAX, BCL2, TP53*, and *CAS3*) [43]. Furthermore, Roy (**1**) and DiRoy (**3**) were cytotoxic against human CCRF-CEM leukemia cells and A549 lung adenocarcinoma cells by altering pro- and antiapoptotic gene expression and induction of apoptosis [44]. A recent study showed that both the acetonic extract of *P. hadiensis* var. *hadiensis* leaves and its major compound, Roy (**1**)**,** were active against the aggressive triple-negative breast cancer cell line, MDA-MB-231S, with IC_50_ values of 25.6 μg/mL and 2.15 μg/mL, respectively [45].

Having the remarkable anticancer activity of this plant in mind, we performed bioactivity-guided isolation of the acetonic extract of *P. hadiensis* var. *hadiensis*. stems, quantified the isolated compounds, and assessed their antiproliferative/cytotoxic effects in GB cell lines.

## 2. Results and Discussion

In the present investigation, we focused on the acetonic stem extract of *P. hadiensis* since results from leaf extracts have been previously published [45].

### 2.1. Phytochemical Study of Plectranthus hadiensis Stems

*P. hadiensis* stems (1.507 kg) were extracted with acetone five times until exhaustion using an ultrasound-assisted extraction method. At the end, a residue of 46.07 g was obtained resulting in a final extraction yield of 2.96% (weight of extract (g)/weight of dry plant (g)). After extraction, several fractionations were carried out by dry flash column chromatography yielding six fractions (I-VI). Subsequently, a qualitative analysis of the fraction composition of the acetonic leaf extract [45], acetonic stem extract, and fractions I-VI from the stems was carried out by using high-performance liquid chromatography coupled to a diode-array detector (HPLC-DAD). The samples were either injected alone or authentic standards (Roy and DiRoy) known to be present in the plant were coinjected [41].

As previously published [45], the *P. hadiensis* leaf extract (Figure 2A) contained a major peak A (Rt: 20.760 min) that was identified as Roy (**1**) compared with previously published ultraviolet (UV) data (218 nm, 271 nm, 410 nm) [46]. Roy (**1**) was coinjected into the HPLC-DAD and coapplied on thin-layer chromatography (TLC).

On the other hand, the acetonic *P. hadiensis* var. *hadiensis* stem extract (Figure 2B) contained three major peaks B (λ = 240, 280, and 420 nm); Rt: 13.304), C (λ = 250, 280, and 350 nm; Rt: 15.119 min) and D (λ = 280 and 420 nm; Rt: 12.175 min). Peak D (λ = 280 and 420 nm) was identified as DiRoy (**3**) compared with previously published UV data (272 nm and 420 nm) [41]). Furthermore, HPLC coinjection and TLC coapplication revealed similar results, i.e., the presence of DiRoy (**3**). The peak A, Roy (**1**)**,** revealed at Rt = 18.926, was also coinjected and coapplied with an authentic sample.

Compounds from fractions III and IV were isolated using liquid column chromatography and preparative thin-layer chromatography (PTLC). In total, six compounds were isolated but due to the scarce quantities only two of them, i.e., Roy (**1**) and DiRoy (**3**), could have been characterized and quantified by NMR, FTIR, melting point and UV spectroscopy (Appendix A).

The HPLC-DAD quantification of the major compounds in *P. hadiensis* leaves, stems, and stem fractions was also performed based on the calibration curves for Roy (**1**) and DiRoy (**3**) (Table 1).

The quantitative analysis by HPLC-DAD (Table 2) showed a huge difference in the amounts of Roy (**1**) between the extract of the leaves and the extract of the stems (5.37 mg/g versus 0.40 mg/g) which was 13.4 times higher for the latter. The extract from the stems contained double the amount of DiRoy (**3**) compared to the leaves (2.15 mg/g versus 1.12 mg/g).

Roy (**1**) and DiRoy (**3**) were both present in fractions II-V. The fraction richest in Roy (**1**) was fraction V (75.68 mg/g), while the richest in DiRoy (**3**) was fraction III (50.33 mg/g). Fractions III and V showed the highest antioxidant and antimicrobial activity (see Section 3.1). Fraction I only contained small quantities of Roy (**1**) (1.03 mg/g), while none of the compounds were detected in fraction VI. All fractions were subjected to antioxidant, general toxicity, and antimicrobial assays (Section 2.3.1).

After the isolation process, the major peaks B (240, 280, and 420 nm; Rt: 13.304) and C (λ = 250, 280, and 350 nm; Rt: 15.119 min) from the acetonic *P. hadiensis* var. *hadiensis* stem extract could not be identified.

### 2.2. BODIPY Synthesis

To synthesize a fluorescent derivative of the abietane diterpenes present in *P. hadiensis*, we focused on the use of 4,4-difluoro-4-bora-3a,4a-diaza-*s*-indacene (BODIPY) as a fluorescent reporter. The BODIPY skeleton is widely applied as a fluorescent marker due to its small size, high extinction coefficient, and fluorescent quantum yield [47]. Therefore, we envisioned developing BODIPY-7α-acetoxy-6β-hydroxyroyleanone derivative (**12**) (Figure 3), which was generated via ester bond formation between the succinimidyl-modified BODIPY (**13**) and 7α-acetoxy-6β-hydroxyroyleanone (**2**).

The synthesis of the BODIPY derivative (**13**) started with the base-catalyzed reaction of succinic anhydride (**15**) with *ortho*-hydroxymethylphenyl-BODIPY (**14**) [48]. Having BODIPY (**13**) successfully prepared, the conjugated BODIPY-abietane (**12**) was obtained using a regioselective coupling mediated by EDC (1-ethyl-3-(3-dimethylaminopropyl) carbodiimide) and 4-DMAP (4-dimethylaminopyridine).

### 2.3. Bioactivity Assays

#### 2.3.1. Studies on the Antioxidant and Antimicrobial Activity and Toxicity of the Fractions

Fractions III and V showed the highest antioxidant and antimicrobial activities, which could be explained by the presence of Roy (**1**) and DiRoy (**3**) (Section 2.1). The antimicrobial activities of fractions III and V were even higher than those of the positive control. None of the samples showed general toxicity in the *Artemia salina* assay (Table 3; Figure 4).

#### 2.3.2. Assessment of the Cytotoxic Effect of *Plectranthus hadiensis* var. *hadiensis* Extracts and Phytochemicals towards GB Cell Lines

The cytotoxic/antiproliferative potential of *P. hadiensis* extracts and phytochemicals, i.e., the extracts from stems and leaves, fractions of the extract (I, II, III, IV, V and VI), and isolated compounds (Roy (**1**) and DiRoy (**3**)) was assessed using the Alamar blue assay. A panel of five glioblastoma cell lines was treated for 24, 48, and 72 h with a serial dilution range of concentrations (100, 50, 25, 6.25, 1.5625, and 0.390625 µg/mL) of these *P. hadiensis* products. Even at the lowest concentrations, Roy (**1**) induced the most pronounced antiproliferative/cytotoxic effect across all cell lines (Appendix A). In fact, DiRoy (**3**) did not significantly affect cell viability once the proliferative capability of cells dropped below 50% relative to the control, except for the results in the U118 cell line (most sensitive), in which treatment with the highest concentrations led to a decrease of 60% in cell viability. The significant difference in the bioactivity of Roy (**1**) and DiRoy (**3**) may be related to the chemical structure of these compounds and their physicochemical properties, such as their molecule polarity. Based on the calculated Log P of the two structures (1.08 for Roy (**1**) and 0.85 for DiRoy (**3**)), it is reasonable to hypothesize that Roy (**1**) could pass the membrane, and reach its target, more easily than DiRoy (**3**), even if the exact mechanism of action of these molecules in glioblastoma cell lines and their organelles is currently under study. Nonetheless, the treatment with fraction II and V exhibited a greater cytotoxic effect against GB cells as these fractions have a higher quantity of DiRoy (**3**) in their chemical composition. Furthermore, treatment with both *P. hadiensis* extracts obtained either from the stems or leaves did not affect the proliferative capacity of brain tumor cells with the exception of the highest concentration (100 µg/mL), which was probably due to the mixture of compounds that masks the bioactivity of certain compounds alone such as Roy (**1**).

IC_50_ values (inhibitory concentration needed to reduce cell viability by 50%) were also calculated for Roy (**1**) and DiRoy (**3**) (Table 4). The lower IC_50_ values were presented for Roy (**1**)**.**

Therefore, all results indicate that Roy (**1**) is the most promising phytochemical to be further explored and as a lead compound to further develop new therapeutic strategies against GB.

#### 2.3.3. Biodistribution in U87 Cells of Roy Labelled with BODIPY

To complement the results obtained for the cell viability assay, Roy (**1**), which had the highest antiproliferative/cytotoxic effect against GB cells, was conjugated with a fluorescent probe to assess its biodistribution in the cells. Using fluorescence and confocal microscopy and acquired data and image analysis, we observed a pronounced fluorescence in the cells treated with both concentrations of Roy-BODIPY (**12**), 6.25 µg/mL (Figure 5B,D) and 25 µg/mL (Figure 5A,C). Moreover, Roy-BODIPY (**12**) accumulated in the cytoplasm of cells instead of the nucleus (pointed out with *N* in each image) (Figure 5). Using confocal microscopy, a distinctive accumulation in cytoplasmic vesicles was visible (marked with an arrow) (Figure 5C,D). Therefore, the increased uptake of Roy (**1**) in cytoplasmic vesicles supported the finding of the higher antiproliferative/cytotoxic activity of this phytochemical (Figure 5). Furthermore, a possible explanation for these results could be the fact that this compound may target the mitochondria and induce the mitochondria-dependent intrinsic pathway of apoptosis. Further studies should be performed to unveil the exact mechanism of action of Roy (**1**).

## 3. Materials and Methods

### 3.1. Plant Material

*Plectranthus hadiensis* (Forssk.) Schweinf. *ex* Sprenger. var. *hadiensis* was grown in *Parque Botânico da Tapada da Ajuda* (Lisbon, Portugal) provided by the Kirstenbosch National Botanical Gardens (Kistenbosch, South Africa). It was collected between June and September in 2007 and 2008. Voucher specimens 833/2007 (11 October 2007) and 438/2010 (25 September 2009) were deposited in the Herbarium *João de Carvalho e Vasconcellos* of the *Instituto Superior de Agronomia* (Lisbon, Portugal). The plant name was checked at http://www.worldfloraonline.org/ (World Flora Online, accessed 12 January 2022). After collecting, the plant was air dried (25 °C) under the shade and stored in cardboard boxes protected from light and humidity to maintain its stability. Leaves and stems were separated at the beginning of this study. Experimental results from the leaves were previously published [45].

### 3.2. Chemicals

All chemicals and reagents used, including the solvents, were of analytical grade. The solvents were distilled before use. Acetone (Ref.: 20066.330), methanol (MeOH, Ref.: 20903.368), sulfuric acid (H_2_SO_4_, Ref.: 470302.872), dichlorometane (DCM, Ref.: 23367.321), and dimethyl sulfoxide (DMSO, Ref.: 23486.322) were from VWR Chemicals). Other materials were hexane (Hex, Fisher Scientific, Ref.: H/0355/21), ethyl acetate (EtOAc; LabChem, Ref.: MP013-9025), trifluoroacetic acid (Sigma-Aldrich, Ref.: 8082600101), sand pure 40–100 mesh (Acrōs organics, Ref.: 370942500); MeOH HPLC grade (Fisher Scientific, Ref.: M/4056/17); ACN HPLC grade (Honeywell, Ref.: 608-001-00-3), K_2_Cr_2_O_7_ (Labonal S.A., Ref.: 2635), and bidistillated water. Furthermore, filter paper (Whatman No. 5 paper, Inc., Clifton, NJ, USA), deuterated-chloroform 99.8 + 0.03% TMS (CDCl_3_, Deutero.de-Rotoquímica, Ref.: 00405), and deuterated methanol (CD_3_OD, Sigma Aldrich, Ref.: 1.06028) were used.

### 3.3. Reagents for Biological Assays

For biological experiments, we used Mueller–Hinton Agar (VWR Chemicals, Prolabo, Ref.: 84686.0500), *Staphylococcus aureus* (Sa, CIP: ATCC 25923), vancomycin (Alfa Aesar, Ref.: J62790), DPPH (Sigma-Aldrich, Ref.: D9132), quercetin (Sigma-Aldrich, Ref.: Q4951), *Artemia* brine shrimp salt (JBL, Ref. 3090600), and *Artemia* brine shrimp eggs (Artemio Mix, JBL, Ref.: 3090600).

The cell lines were maintained with Dulbecco’s Modified Eagle’s Medium-high glucose (DMEM-HG) (Biowest, Nuaillé, France) supplemented with 10% (*v*/*v*) of heat-inactivated fetal bovine serum (FBS) (Sigma, St. Louis, MO, USA) and 1% (*v*/*v*) of penicillin-streptomycin (Sigma, St. Louis, MO, USA), stock solution of Resazurin salt dye at a concentration of 0.1 mg/mL.

### 3.4. General Equipment

The equipment consisted of a grinder (FLAMA 1705 FL 700 W 230 V~50 Hz), ultrasound apparatus (VWR USC300-TH; VWR, Lutterworth, United Kingdom), rotary evaporator (Buchi Rotavapor R-100), UV lamp (CAMAG, Ref.: 022.9230), microscopy (OE, CETI Belgium), and an air pump (HI-FLO^TM^ Single Type 4000).

### 3.5. General Experimental Procedures

#### 3.5.1. Extract Preparation

Dried and powdered *P. hadiensis* var. *hadiensis* stems (1.507 kg) were extracted until exhaustion with acetone (31.95 L × 5, 30 min each) at room temperature for 2 weeks using an ultrasound-assisted extraction method [49]. The ultrasonic bath was operated at 35 Hz with a maximum input power of 320 W. The extract was filtered using Whatman No. 5 paper and the filtrate was concentrated using a rotary evaporator under reduced pressure at low temperature (40 °C). The extraction yield (% *w/w*) was determined by Equation (1):(1)$$\%Yield\;(w/w)=\frac{Weight\;of\;extract\;\left(g\right)}{Weight\;of\;dried\;plant\;\left(g\right)}\times 100$$

#### 3.5.2. Bioassay-Guided Fractionation

The extract residue was subjected to chromatographic fractionation. Dry flash column chromatography was carried out with silica gel 60 (0.063–0.200 nm) for column chromatography (Ref.: 1.007734.1000, Merck) using eluents Hex:EtOAc 80:20 and 85:15 following the procedure and silica quantities described in [50,51] and collecting the fractions in test tubes of 70 mL. It was performed with a funnel of 10 cm diameter × 8.5 cm length with a porous filter (Pyrex France, Fabrication Sorivel France No. 3), and another funnel of 10 cm diameter × 9 cm length with a porous filter (Pyrex France, Fabrication Sorivel France No. 2).

TLC was used to monitor the fractionation performed according to the method described in the WHO guidelines with modifications [52]. Briefly, a concentrated filtrate was dissolved in the minimum quantity of hexane or acetone and 5 μL of extract was spotted in silica gel 60 F_254_ aluminum plates (Ref.: 1.05554.0001; Merck). Plates were developed in hexane/ethyl acetate (80:20, 85:15, and 70:30) and in DCM/Acetone (99:1). The spots were observed at 254 and 366 nm in a UV lamp while spraying with methanol-sulfuric acid.

#### 3.5.3. Isolation and Chemical Characterization Procedure

Flash liquid column chromatography was performed to obtain isolated compounds from stem fractions following previously described procedures [53,54,55]. A glass column filled with silica gel 60 (0.040–0.063 nm) was used for column chromatography (Ref.: 1.09385.1000, Merck; Ref.: 27623.323, VWR Chemicals), and 20 mL test tubes were used to collect the fractions. The vacuum system consisted of a 90° glass adaptor linked to an aquarium air pump as a pressure source. This assembly can be inserted into the column and clamped in place. Flash liquid column chromatography was performed with two columns; the columns (one of 4 cm diameter × 46 cm height × 500 mL capacity and another one of 2.5 cm diameter × 66 cm height × 200 mL capacity) were purchased from Normax (Marinha Grande, Portugal).

TLC was used to monitor this process and to detect the compounds which were compared to standards (Roy(**1**) and DiRoy(**3**)) as explained in Section 2.1. To purify and isolate compounds from the obtained fractions, homemade PTLC plaques containing silica gel 60 for PTLC (Ref.: 1.07747.1000, Merck) were used following the procedure indicated in [56]. Briefly, the sample was dissolved in the minimum volume of solvent possible. Then, the sample was applied to the bottom of a homemade preparative silica UV indicator plate (1.5 cm from the bottom) as a thin straight line (2–4 mm) using a Pasteur pipette. The plate was run with 100 mL of freshly prepared mobile phase previously chosen. When the solvent front was a few milliliters from the top, the plate was removed from the tank and air dried in the fume cupboard. Next, UV-absorbing bands were marked out with a pencil, scraped off onto tin foil, and put into a sintered glass funnel attached to a glass Buchner flask to which vacuum was applied. The filter removed the silica and the solution with the compound was evaporated on a rotary evaporator. Finally, isolated compounds were confirmed by TLC coapplication with authentic standards (Roy, DiRoy) before characterizing their structure.

#### 3.5.4. BODIPY-7α-acetoxy-6β-hydroxyroyleanone derivative (12) Synthesis

Succinic anhydride (42 mg, 0.42 mmol) was added to a mixture of BODIPY (**14**) (100 mg, 0.28 mmol) and triethylamine (79 μL, 0.56 mmol) in anhydrous dichloromethane (4 mL). The reaction mixture was stirred at room temperature until complete consumption of the starting material was observed by TLC. The solution was diluted with DCM and aqueous HCl 1M was added until pH 3. The resulting mixture was washed with H_2_O, the layers were separated, and the organic layer was washed with brine, dried, and concentrated. The crude material of (**13**) was used in the next step without further purification (112 mg, 88%). ^1^H NMR (300 MHz, CDCl_3_) δ 7.48 (td, *J* = 2.4, 1.4 Hz, 1H), 7.26 (d, *J* = 0.7 Hz, 0H), 5.98 (d, *J* = 1.2 Hz, 1H), 5.05 (d, *J* = 1.4 Hz, 1H), 2.54 (d, *J* = 1.3 Hz, 4H), 1.35 (s, 3H).

A mixture of carboxylic acid (**13**) (27 mg, 0.06 mmol), and abietane (**1**) (10 mg, 0.03 mmol) in anhydrous dichloromethane (2 mL) was cooled to 0 °C and treated under argon atmosphere with EDC (1-[3-(dimethylamino)propyl]-3-ethylcarbodiimide methiodide) (18 mg, 0.06 mmol) and a catalytic amount of DMAP (3 mg). The reaction mixture was stirred for 90 min. The solution was diluted with DCM and washed with H_2_O. The layers were separated, and the organic layer was dried and concentrated. The crude material was purified through silica column chromatography (hexane/ethyl acetate 85:15) to yield compound (**12**) (16 mg, 65%). ^1^H NMR (500 MHz, CDCl_3_) δ 7.55–7.37 (m, 3H), 7.31–7.23 (m, 2H), 6.00 (bs, 2H), 5.65 (dd, *J* = 2.1, 0.6 Hz, 1H), 5.08 (d, *J* = 1.4 Hz, 2H), 4.32 (s, 1H), 3.10 (p, *J* = 7.1 Hz, 1H), 2.87 (t, *J* = 7.0 Hz, 2H), 2.76–2.58 (m, 2H), 2.18 (s, 3H), 2.05 (s, 3H), 1.62 (s, 3H), 1.37 (d, *J* = 2.7 Hz, 7H), 1.23 (s, 3H), 1.17 (dd, *J* = 7.1, 4.8 Hz, 6H), 0.95 (s, 3H); ^13^C NMR (126 MHz, CDCl_3_) δ 185.7, 179.5, 171.2, 169.7, 169.5, 155.9, 152.8, 149.2, 142.8, 139.5, 138.9, 135.7, 134.1, 133.4, 131.0, 129.6, 129.5, 129.3, 129.2, 129.1, 129.1, 128.6, 128.5, 121.4, 77.3, 77.0, 76.8, 68.8, 67.2, 64.2, 63.9, 49.8, 42.3, 38.9, 38.3, 33.7, 33.5, 31.9, 30.9, 29.7, 29.4, 28.6, 28.5, 28.5, 25.2, 23.9, 22.7, 21.7, 20.9, 20.2, 20.2, 19.1, 18.9, 17.7, 14.6, 14.6, 14.2, 14.1, 14.0, 13.9. HRMS (ESI) [M+Na]^+^ *m*/*z* calcd for C_46_H_53_BF_2_N_2_NaO_9_ 849.3710.9294, found 849.3733.

#### 3.5.5. Compound Characterization

##### Nuclear Magnetic Resonance (NMR)

^1^H-NMR and ^13^C-NMR analyses were carried out in CDCl_3_ and CD_3_OD at room temperature on a Bruker^®^ Biospin Fourier spectrometer (^1^H-NMR and ^13^C-NMR spectra were recorded at 300 MHz and 75 MHz, respectively) operating at 315K at the Faculty of Pharmacy (University of Lisbon, Lisbon, Portugal) and on a Bruker^®^ Advance 400 spectrophotometer (^1^H-NMR and ^13^C-NMR spectra were recorded at 400.13 MHz and 100.61 MHz, respectively) operating at 293K at the Faculty of Sciences (University of Lisbon).

The chemical shifts were given in parts per million (ppm, δ) and referenced to the CDCl_3_ peaks at δ = 7.26 ppm (^1^H-NMR) and δ = 77.16 ppm (^13^C-NMR) and to the CD_3_OD peaks at δ = 3.31 ppm (^1^H-NMR) and δ = 49.00 ppm (^13^C-NMR).

##### Fourier Transform Infrared Spectroscopy (FTIR)

Infrared spectra were obtained using the PerkinElmer^®^ Spectrum TWO 400 (instrument number: L1600400; PerkinElmer Inc, Waltham, MA, USA) equipped with an attenuated total reflectance (ATR) device.

The samples were evaluated by FTIR in a PerkinElmer^®^ Spectrum TWO 400 (PerkinElmer Inc, Waltham, MA, USA) equipped with an attenuated total reflectance (ATR) device. The ATR system was cleaned before each analysis using dry paper and scrubbing it with methanol and water (50:50). The air in the analysis room of the FTIR-ATR spectrum was used as background to verify the cleanliness and to evaluate the instrumental conditions and room interferences due to H_2_O and CO_2_. The spectra were obtained collecting four scans of each sample, between 4000 and 600 cm^−^^1^ with a resolution of 4 cm^−^^1^.

#### 3.5.6. HPLC-DAD

##### Methodology

The extracts and fraction profiling were performed with an Agilent Technologies 1200 Infinity Series HPLC system equipped with a manual injector (G 1328C), vail sampler (G 7129A), quaternary pump (G 1311C), thermostat (G 7116A), a column oven eclipse XBD-C18 80 Å (4.6 × 250 mm, 5 μm), and a diode-array detector (G 1315D). Data processing was handled by Agilent OpenLAB CDS ChemStaton Workstation, revision C.01.01.

##### Qualitative and Quantitative Analysis

Four detection wavelengths were selected: 254, 270, 280, and 330 nm. A 20 μL sample was injected into a reverse phase Agilent Eclipse XDB-C18 5 μm (4.6 × 250 mm) 80 Å column and eluted with a gradient composed of methanol (A), acetonitrile (B), and 0.3% (*w*/*v*) trifluoroacetic acid in ultrapure water (C). The employed method was adapted from the one previously published [57] as follows: 0 min, 15% A, 5% B, and 80% C; 10 min, 70% A, 30% B, and 0% C; 25 min, 70% A, 30% B, and 0% C; and 28 min, 15% A, 5% B, and 80% C. The flow rate was set at 1 mL/min at 29 °C temperature. Solvents were previously filtered and degassed through a 0.22 μm membrane filter.

The component yield for Roy and DiRoy in the extracts was determined as follows: samples of the extracts (or fractions) were injected at 10 mg/mL. The resulting chromatograms, showing the presence of several products with different Rt and areas, were analyzed. The presence of the compounds of interest was confirmed by doing a coinjection of the extract and the pure compound. UV spectra and Rt, together with the coinjection outcome, allowed us to identify which peak of the sample corresponded to Roy and DiRoy. The respective areas were replaced to the X value on their calibration curves. The obtained value indicated the amount of Roy (or DiRoy) in 10 mg/mL of the sample. Next, the quantity of Roy (or DiRoy) in 1 mg/mL of the sample was calculated, dividing by 10. Finally, the quantity of Roy (or DiRoy) in 1 g/mL of the sample was obtained by multiplying it by 1000.

#### 3.5.7. Bioactivity Assays

##### Determination of Antioxidant Activity (DPPH)

The antioxidant activity of the fractions was measured by the DPPH method as described in [58]. Briefly, 10 μL of each sample was added to a 990 μL solution of DPPH (0.002% in methanol). The mixture was incubated for 30 min at room temperature. The absorbance was measured at 517 nm against a corresponding blank and the antioxidant activity was calculated using Equation (2):(2)$$\mathrm{AA}\%=\frac{\left(A_{\mathrm{DDPH}}-A_{\mathrm{Sample}}\right)}{A_{\mathrm{DPPH}}}\times 100$$

AA was the percentage of antioxidant activity, A_DPPH_ was the absorption of DPPH against the blank, and A_Sample_ was the absorption of the sample or control against the blank. Tests were carried out in triplicate at a sample concentration at 1 mg of sample/mL. The reference standard used for this procedure was quercetin, used in the same conditions as the samples.

##### Antibacterial Activity: Well Diffusion Method

The antibacterial activity of the fractions was evaluated against Gram-positive *S. aureus* [32]. The samples were reconstituted in DMSO at a concentration of 1 mg/mL. Stock solutions of vancomycin as the reference antibiotic were also prepared at 1 mg/mL DMSO.

In aseptic conditions, Petri dishes containing 20 mL of solid Mueller–Hinton for bacteria were inoculated with 0.1 mL of bacterial suspension matching a 0.5 McFarland standard solution and uniformly spread on the medium surface using a sterile swab. Wells of approximately 5 mm in diameter were made in the medium, using a sterile glass Pasteur pipette and 50 μL of each extract was added into the wells. Vancomycin as positive control against Gram-positive bacteria and DMSO as negative control were used in the assay. Plates were incubated at 37 °C for 24 h. The antimicrobial activity was evaluated by measuring the diameter (in mm) of the inhibition zone formed around the wells and compared to the controls.

##### Brine Shrimp Lethality Bioassay

In order to evaluate the general toxicity of the different samples, a lethality test using *Artemia salina* brine shrimp was performed as described [59]. Concentrations of 10 ppm of each fraction were tested. The number of dead larvae was recorded after 24 h and used to calculate the lethal concentration (% dead animals), according to Equation (3):(3)$$Lethal\;Concentration\;\left(dead\%\right)=\frac{\left(Total\;A.salina-Alive\;A.\;salina\right)}{Total\;A.\;salina}\times 100$$

##### In Vitro Assays

Cell Culture

The brain tumor cell lines were: U87 (brain-like glioblastoma), A172 (glioblastoma), H4 (neuroglioma), U118 (astrocytoma/glioblastoma), and U373 (astrocytoma/glioblastoma) cell lines. U118 and U373 cell lines were a kind gift from Prof. Maria Conceição Pedroso de Lima (Faculty of Sciences and Technology, Center for Neuroscience and Cell Biology, University of Coimbra, Coimbra, Portugal) and U87, A172, and H4 cell lines were kindly provided by Prof. Carla Vitorino (Faculty of Pharmacy, University of Coimbra, Coimbra, Portugal). Cells were cultivated in Dulbecco’s Modified Eagle’s Medium-high glucose (DMEM-HG) (Biowest, Nuaillé, France) supplemented with 10% (*v/v*) of heat-inactivated fetal bovine serum (FBS) (Sigma, St. Louis, MO, USA) and 1% (*v/v*) of penicillin-streptomycin (Sigma, St. Louis, MO, USA) and maintained at 37 °C and 5% CO_2_. For experiments, cells were used at 80% confluence.

2.Cell Viability

U87, A172, U118, U373, and H4 cells (1 × 10^4^ cells/well) were seeded in 96-well plates, 24 h before treatment. Cells were treated with the same concentrations of the extract from *P. hadiensis* leaves, the extract from *P. hadiensis* stems, the fractions of the extracts (I, II, III, IV, V, VI), and the isolated compounds (Roy (**1**) and DiRoy (**3**)), or with the vehicle control (DMSO), and further incubated for 24, 48, and 72 h. Cell viability was assessed using a modified Alamar blue assay [60]. Briefly, a solution of DMEM-HG medium with 10% (*v/v*) of a stock solution of resazurin salt dye at a concentration of 0.1 mg/mL was prepared, which was further added to each well after 24, 48, and 72 h. After a 4 h incubation at 37 °C and 5% CO_2_, the absorbance of the plate was read at 570 and 600 nm in a BioTek reader (BioTek Instruments, Inc., Winooski, VT, USA). The absorbance results were obtained by the Gen5 program. Cell viability was then calculated in accordance with Equation (4):(4)$$Cell\;viability\;\left(\%\right)=\frac{\left(A570-A600\right)\;of\;treated\;cells}{\left(A570-A600\right)\;of\;control\;cells\;}\times 100\%$$

Half-maximal inhibitory concentration (IC_50_) values were further calculated using GraphPad Prism Software v.7.04 (GraphPad Software Inc., San Diego, CA, USA) [61].

3.Fluorescence Imaging

U87 cells (1 ×10^5^ cells/well) were seeded in µ-Dish 35 mm high (ibidi, Germany). After 24 h, cells were treated with different concentrations of BODIPY-7α-acetoxy-6β-hydroxyroyleanone derivative (**12**) (6.25 and 25 µg/mL) and further incubated for 48 h (5% CO_2_ and 37 °C). Cells were then washed twice with phosphate-buffered saline solution (PBS) and fixed with 4% paraformaldehyde in PBS for 15 min at room temperature [60]. Images were acquired using a laser scanning confocal inverted microscope (LSM 710 configured to an Axio Observer Z1 microscope (Carl Zeiss, Germany) using a 63× oil objective (Plan-Apochromat, 1.4 NA).

#### 3.5.8. Others

Chemical structures were drawn using ChemDraw 20.1.1. software (Perkin-Elmer Informatics). NMR Spectra were processed by MestRe Nova 12.0.1-20560 and FT-IR spectra by Spectrum IR (10.6.1) and Spectrum Quant PerkinErmer Software (10.6.1.942).

## 4. Conclusions

We reported on the bioguided isolation of compounds from the acetonic *P. hadiensis* var.* hadiensis* stems extract and the in vitro antiglioblastoma activity of the extract, fractions, and isolated constituents. Leaves and stems samples from *P. hadiensis* var. *hadiensis* were extracted five times using acetone in an ultrasound-assisted extraction method. Roy (**1**) and DiRoy (**3**) compounds were isolated from the *P. hadiensis* var. *hadiensis* stem extract.

By using a HPLC-DAD analysis, we found significative differences between extracts from leaves compared with those from stems, mainly in their content of Roy (**1**). This was verified by TLC (authentic sample) and HPLC-DAD. Roy (**1**) was mainly present in the leaves while the stems were richer in DiRoy (**3**). Moreover, this is the first paper that provides a HPLC-DAD quantification of *P. hadiensis* compounds in acetonic extracts of the leaves and stems.

The activity of the pure compound Roy (**1**) was higher than any of the activities found in extracts, fractions, and the compound DiRoy (**3**)**,** even at the lowest concentrations across all the tested cell lines (U87, A172, H4, U118, and U373).

The significant difference in the bioactivity of Roy (**1**) and DiRoy (**3**) may be related to the chemical structure of these compounds and their physicochemical properties, such as their molecule polarity. The fractions V and II showed the highest activity in cell lines among the fractions tested. Moreover, fractions III and V showed the highest antioxidant and antimicrobial activities. The higher values shown in the biological assays in fraction V can be explained by the fact that this fraction was the richest in Roy (**1**).

The increase of cellular fluorescence by the fluorescent BODIPY-labeled Roy (**1**), i.e., BODIPY-7α-acetoxy-6β-hydroxyroyleanone (**12**) supported the finding of the higher antiproliferative effects of Roy (**1**) in the cell assays.

Taken together, the results indicate that Roy (**1**) is a promising natural compound that may serve as a lead compound to generate semisynthetic derivatives with improved activity against GB.

## Figures and Tables

**Figure 1 molecules-27-03813-f001:**
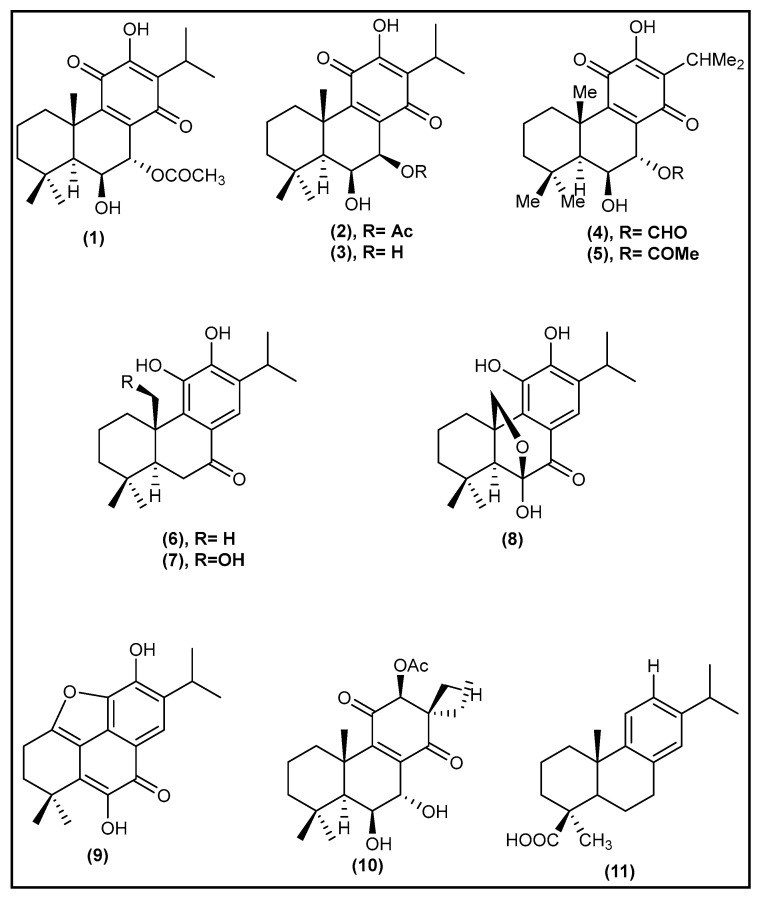
Chemical structures of 7α-acetoxy-6β-hydroxyroyleanone (Roy,**1**), its isomer 7β-acetoxy-6β-hydroxyroyleanone (**2**), 6β,7β-dihydroxyroyleanone (DiRoy,**3**), 7-formyloxy-6ß,12-dihydroxy-abieta-8,12-diene-11,14-dione (**4**), 7α-acetoxy-6ß,12-dihydroxy-abieta-8,12-diene-11,14-dione (**5**), 11-hydroxysugiol (**6**), 11,20-dihydroxysugiol (**7**), carnosolon (**8**), 1,11-Epoxy-6,12-dihydroxy-20-norabieta-1(10),5,8,11,13-pentaen-7-one (**9**), coleone P (**10**), and callistric acid (**11**).

**Figure 2 molecules-27-03813-f002:**
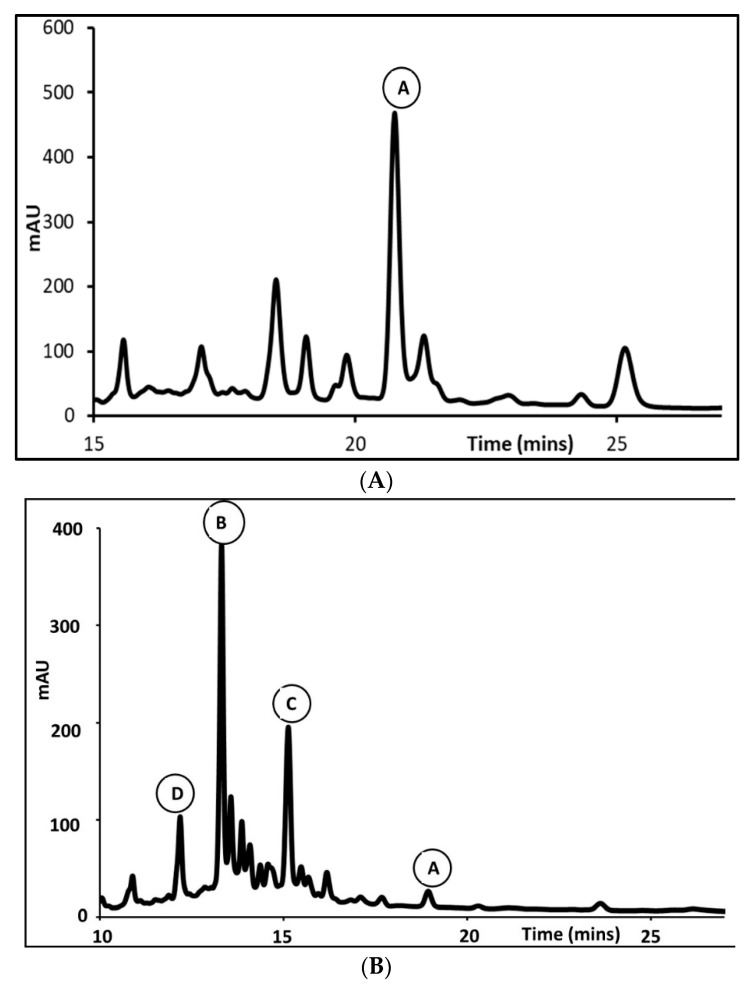
HPLC profile chromatograms (270 nm) of *P. hadiensis* var. *hadiensis* leaf (**A**) and stem (**B**) acetone extracts. Peak A: Roy (1); Peak B and Peak C are major compounds that appeared on the HPLC-DAD chromatogram of the *P. hadiensis* stems extract that could not been structurally characterized in the present study; Peak D: DiRoy (3).

**Figure 3 molecules-27-03813-f003:**
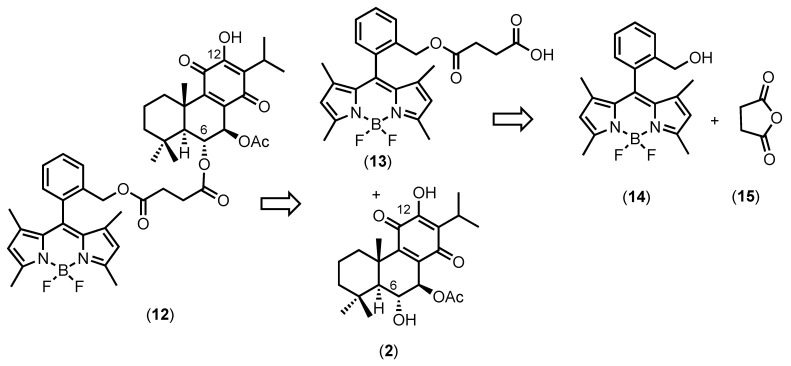
Retrosynthesis for the preparation of BODIPY-7α-acetoxy-6β hydroxyroyleanone derivative (**12**).

**Figure 4 molecules-27-03813-f004:**
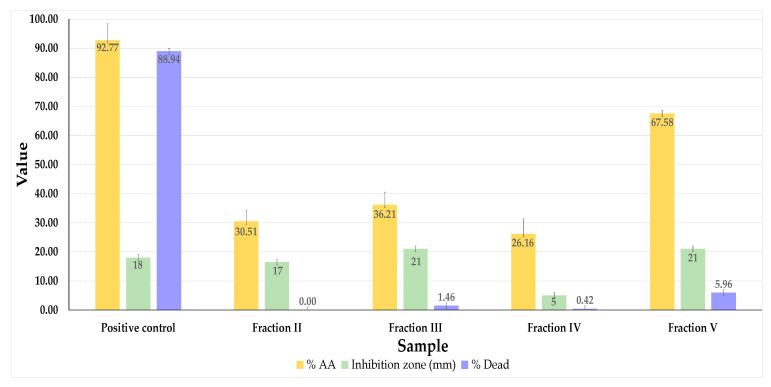
Antioxidant activity (expressed in % AA), antibacterial activity against Gram-positive *S. aureus* (expressed in inhibition zone diameter, mm), and general toxicity (expressed in % dead *A. salina*) of fractions II-V. AA, antioxidant activity. Positive controls: antioxidant activity (quercetin); mortality (potassium dichromate); and antimicrobial activity (vancomycin).

**Figure 5 molecules-27-03813-f005:**
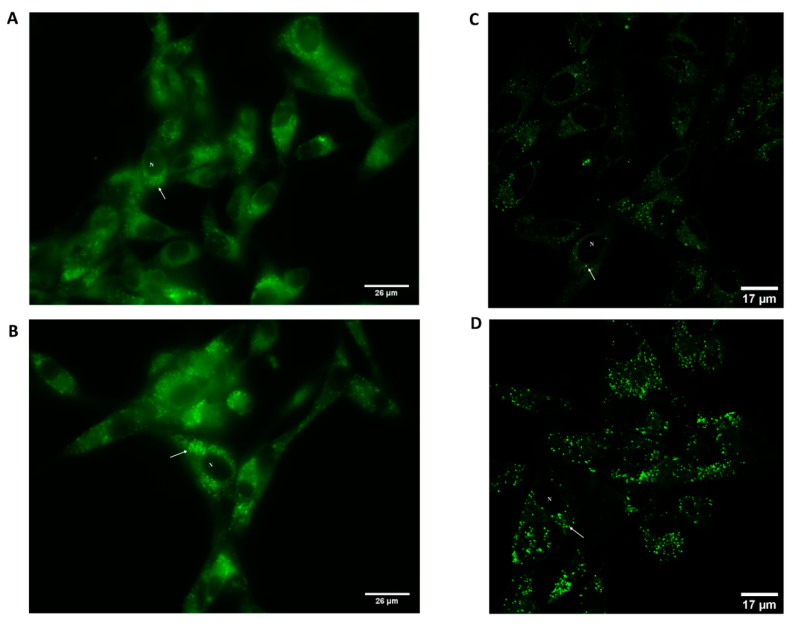
Biodistribution of Roy (**1**) isolated from *Plectranthus hadiensis* var. *hadiensis* extract in U87 cells. Cells were treated with 6.25 µg/mL (**A**,**C**) and 25 µg/mL (**B**,**D**) of Roy-BODIPY for 48 h. Representative images of fluorescence microscopic images (**A**,**B**) and confocal laser scanning microscopic images (**C**,**D**) show Roy-BODIPY (green fluorescence) uptake by U87 cells (Scale bar 26 and 17 µm, respectively). Note: The cell nucleus is represented with an *N* and the vesicles are pointed out with a representative arrow.

**Table 1 molecules-27-03813-t001:** Quantitative analysis of compounds (**1**) and (**3**) in *Plectranthus hadiensis* extracts and fractions using HPLC–DAD.

Compounds	Linear Regression Data		LOD	LOQ
	Calibration Curve	R^2^		
Roy (**1**)	y = 29,435x + 91,338	0.9984	0.0009	0.0027
DiRoy (**3**)	y = 50,426x − 43,699	0.9995	0.00004	0.0001

**Table 2 molecules-27-03813-t002:** Component quantification of *P. hadiensis* samples.

Sample	Component Yield in Extract (mg/g)	
	Roy (1)	DiRoy (3)
*Plectranthus hadiensis* var. *hadiensis* leaves	5.37	1.12
*Plectranthus hadiensis* var. *hadiensis* stems	0.40	2.15
Fraction I	1.03	n/d
Fraction II	11.8	3.81
Fraction III	8.77	50.03
Fraction IV	2.15	9.50
Fraction V	75.68	1.84
Fraction VI	n/d	n/d

n/d: not detected. The component yield is expressed in mg of component per g of plant dry material.

**Table 3 molecules-27-03813-t003:** Antioxidant activity, antimicrobial activity against Gram-positive *S. aureus,* and toxicity towards *Artemia salina*.

Sample	% AA	SDV	Inhibition Zone (mm)	SDV	% Dead	SDV
Positive control	92.77	5.61	18	0	88.94	0.07
Fraction II	30.51	3.88	17	1	0.00	0.00
Fraction III	36.21	4.31	21	1	1.46	0.55
Fraction IV	26.16	5.17	5	0	0.42	0.59
Fraction V	67.58	0.95	21	1	5.96	5.29

*p* value < 0.05 was considered to indicate statistical significance. AA, antioxidant activity; SDV, standard deviation. Positive controls: antioxidant activity (quercetin); mortality (potassium dichromate); and antimicrobial activity (vancomycin).

**Table 4 molecules-27-03813-t004:** Cytotoxic effects of Roy and DiRoy isolated from *P. hadiensis* towards five GB cell lines. IC_50_ values of each three independent experiments are shown.

	IC_50_ (µg/mL)					
	U87	A172	U118	U373	H4	
Roy	40.98	46.78	7.64	6.27	56.91	24 h
DiRoy	ND	ND	17.00	57.41	ND	
Roy	23.19	33.05	8.84	6.21	18.09	48 h
DiRoy	ND	ND	5.79	33.7	ND	
Roy	7.111	26.31	8.04	ND	12.99	72 h
DiRoy	ND	ND	12.07	19.30	ND	

## Data Availability

The raw data supporting the conclusions of this article will be made available by the authors, without undue reservation, to any qualified researcher.

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
