# Peer review of "Phytochemical Study and Antiglioblastoma Activity Assessment of Plectranthus hadiensis (Forssk.) Schweinf. ex Sprenger var. hadiensis Stems"

_molecules, 2022, doi:10.3390/molecules27123813_

Round 1

Reviewer 1 Report

In “ Phytochemical study and anti-glioblastoma activity  assessment of Plectranthus hadiensis (Forssk.) Schweinf. ex  Sprenger var. hadiensis stems” Eva María Domínguez-Martín et al.  isolated some compounds from the acetone extract of P. hadiensis stems and investigated the in vitro anti-glioblastoma activity of the extract and its isolated constituents.

The manuscript is interesting and well written, however the authors

·        should cite Figure 4 in the text and add significance and p-value;

·        should add a positive control and a negative control in Figure 5.

Author Response

Comment 1.1.: The authors should cite Figure 4 in the text and add significance and p-value.

Authors: We thank the reviewer for the comment. “Figure 4” was cited in line 290, while “p value < 0.05 was considered to indicate statistical significance” was added in line 298.

Comment 1.2.: The authors should add a positive control and a negative control in Figure 5.

Authors: We thank the referee for the valuable comment and suggestion, although the in vitro results that we presented in this work are preliminary data, in which this assay was only to show that the isolated molecule was able to enter and accumulate inside the cells. We do not have a positive control and the confocal images for the negative control (untreated cells) are black because cells do not have fluorescence. However, for further cellular assays, we will consider this valuable suggestion to improve this part with a positive control and labeling cell organelles for better perception.

Reviewer 2 Report

1. Since that is a script describing a systematic, step-by-step method to extract and purify the Roy and related compounds, presentation of the spectroscopic data like UV, NMR and FTIR is not sufficient.  It is suggested to check the sharpness of melting point for purity of isolated single compound to be established. Otherwise, other components in the extract could lead to the mentioned anti-tumour effects.

2. In Table 1, the R2 value of Roy(1) is found below 0.9990 which does not seem to be sufficiently linear.

3. Is it possible to describe how the "Component yield in extract (mg/g)" for Roy and DiRoy in Table 2 were determined?

4. It is very nice to observe the distribution of Roy inside the cells by Florescent probing using BODIPY. Results from fluorescence and confocal microscopy show some differences in the distribution. Is there any further evidence to indicate attachment of Roy molecules to mitochondria?

5. In paragraph 3.5.3 and 3.5.6, a single narrow UV absorbing band (from HPLC-DAD and TLC elution of a mixture (spiked sample) of an authentic sample of Roy and a purified sample of Roy from the extract) is suggested to definitely help in identification of Roy. Is there any data from the spiked sample?

6. In conclusion, the difference in bioactivity of Roy and DiRoy was explained as "The significant difference in the bioactivity of Roy (1) and DiRoy (3) may be related to the chemical structure of these compounds and their physicochemical properties such as the molecule polarity.". Is there any clue for the type of interaction between Roy and DiRoy, and the targeted tumour cells/organelles?

Author Response

Comment 2.1.: Since that is a script describing a systematic, step-by-step method to extract and purify the Roy and related compounds, presentation of the spectroscopic data like UV, NMR and FTIR is not sufficient.  It is suggested to check the sharpness of melting point for purity of isolated single compound to be established. Otherwise, other components in the extract could lead to the mentioned anti-tumour effects.

Authors: We thank the reviewer for the comment. Unfortunately, we do not have the opportunity to measure the melting point of the pure compounds, in time to re-submit the paper after the revision step. However, we attach the NMR spectra of the isolated compounds Roy (1) and DiRoy (3). As it is possible to notice, there are no trace of impurities, and the signals of the analysed compounds are in agreement with those reported in other previous published works. [1,2]

Comment 2.2.: In Table 1, the R2 value of Roy(1) is found below 0.9990 which does not seem to be sufficiently linear.

Authors: Thanks to the reviewer comment, the calibration curves and the corresponding tables of Roy (1) and DiRoy (3) have been added to Supplementary Material to justify their linearity.

Comment 2.3.: Is it possible to describe how the "Component yield in extract (mg/g)" for Roy and DiRoy in Table 2 were determined?

Authors: Yes, and the following explanation was added in the manuscript (section 3.5.6.2): “The component yield for Roy and DiRoy in the extracts was determined as follows: Samples of the extracts (or fractions) were injected at 10 mg/mL. The resulting chromatograms, showing the presence of several products with different Rt and areas, were analysed. The presence of the compounds of interest was confirmed by doing a co-injection of the extract and the pure compound. UV spectra and Rt, together with the co-injection outcome, allowed us to identify which peak of the sample corresponds to Roy and DiRoy. The respective areas were replaced to the X value on their calibration curves. The obtained value indicated the amount of Roy (or DiRoy) in 10 mg/mL of the sample. Next, the quantity of Roy (or DiRoy) in 1 mg/mL of the sample was calculated, dividing by 10. Finally, the quantity of Roy (or DiRoy) in 1 g/mL of the sample  was obtained multiplying it *1000.”

Comment 2.4.: It is very nice to observe the distribution of Roy inside the cells by Florescent probing using BODIPY. Results from fluorescence and confocal microscopy show some differences in the distribution. Is there any further evidence to indicate attachment of Roy molecules to mitochondria?

Authors: We are grateful for this valuable comment and question. Therefore, the in vitro results that we presented in this work are preliminary data, in which the goal of this assay was to show the biodistribution of Roy inside the cells, as well as, its successful uptake by the cells, supporting the data obtained by the viability studies. Currently, we do not have any further evidence supporting our preliminary conclusion that Roy may be accumulated into the mitochondria, but we have ongoing and future work exploring the molecular and cellular mechanism of action of this compound.

Comment 2.5.: In paragraph 3.5.3 and 3.5.6, a single narrow UV absorbing band (from HPLC-DAD and TLC elution of a mixture (spiked sample) of an authentic sample of Roy and a purified sample of Roy from the extract) is suggested to definitely help in identification of Roy. Is there any data from the spiked sample?

Authors: Yes, in TLC the deposited pure Roy showed an Rf = 0.19 using hexane/ethyl acetate as eluent, as well as one of the spots in the sample. The presence of Roy has been furtherly confirmed by co-application of pure Roy and the sample. In the HPLC-DAD chromatogram, the peak corresponding to Roy in the mixture increased in intensity after co-injection with the authentic sample of Roy.

Comment 2.6.: In conclusion, the difference in bioactivity of Roy and DiRoy was explained as "The significant difference in the bioactivity of Roy (1) and DiRoy (3) may be related to the chemical structure of these compounds and their physicochemical properties such as the molecule polarity.". Is there any clue for the type of interaction between Roy and DiRoy, and the targeted tumour cells/organelles?

Authors: We thank the reviewer for the comment. In order to better explain our statement, the following sentence has been included in the manuscript (line 579-585): “Basing on the calculated LogP of the two structures (1.08 for Roy (1) and 0.85 for DiRoy (3)), it is reasonable to hypothesize that Roy (1) could pass the membrane, and reach its target, more easily than DiRoy (3), even if the exact mechanism of action of these molecules in glioblastoma cell lines and their organelles is currently under study.”

Bibliography

  1. Mehrotra, R.; Vishwakarma, R.A.; Thakur, R.S. Abietane Diterpenoids from Coleus Zeylanicus. Phytochemistry 1989, 28, 3135–3137, doi:https://doi.org/10.1016/0031-9422(89)80293-6.
  2. Ntungwe, E.; Domínguez-Martín, E.M.; Teodósio, C.; Teixidó-Trujillo, S.; Armas Capote, N.; Saraiva, L.; Díaz-Lanza, A.M.; Duarte, N.; Rijo, P. Preliminary Biological Activity Screening of Plectranthus Spp. Extracts for the Search of Anticancer Lead Molecules. Pharmaceuticals 2021, 14.

Reviewer 3 Report

This manuscript could be accepted for publication in Molecules. The topic of this manuscript is novel and important for medicine and medicinal chemistry. Glioblastoma (GB) is the most malignant form of primary astrocytoma, accounting for more than 60% of all brain tumors in adults. Nowadays, due to the development of multidrug resistance causing relapses to the current treatments and the development of severe side effects resulting in reduced survival rates, new therapeutic approaches are needed. The genus Plectranthus belongs to the Lamiaceae family and is known to be rich in abietane-type diterpenes which possess antitumor activity. Specifically, P. hadiensis (Forssk.) Schweinf. ex Sprenger has been documented for the use against brain tumors. Therefore, the aim of this work was to perform the bio-guided isolation of compounds from the acetonic extract of P. hadiensis stems and to investigate the in vitro anti-glioblastoma activity of the extract and its isolated constituents. After extraction, six fractions were obtained from the acetonic extract of P. hadiensis stems. In a preliminary biological screening, the fractions V and III showed the highest antioxidant and antimicrobial activities. None of the fractions were toxic in the Artemia salina assay. Authors yielded different abietane-type diterpenes such as 7α-acetoxy-6β-hidroxyroyleanone (Roy) and 6β,7β-dihydroxyroyleanone (DiRoy) which was also in agreement with the HPLC-DAD profile of the extract. Furthermore, the antiproliferative activity was assessed in a glioma tumor cell line panel by the Alamar blue assay. After 48 h treatment, Roy exerted strong antiproliferative/cytotoxic effects against tumor cells with low IC50 values among the different cell lines. Finally, authors synthesized a new fluorescence derivative in this study to evaluate the biodistribution of Roy. The uptake of BODIPY-7α-acetoxy-6β-hydroxyroyleanone by GB cells was associated with increased intracellular fluorescence supporting the antiproliferative effects of Roy. In conclusion, Roy is a promising natural compound that may serve as a lead compound for further derivatization to develop future therapeutic strategies against GB.

The introduction provide sufficient background (my only suggestion is add in this section some references to works related to synthetic potential antitumor drugs and their precursors: Journal of Molecular Structure 2016 1111, 142-150; New Journal of Chemistry 2017 41 (9), 3246-3250. The research methodology is adequate and novel. The results are clearly presented. The conclusions supported by the data. The manuscript good illustrated and interesting to read. The style and English language is fine. Finally, may be, section 3.5.3. Isolation and chemical characterization procedure could be moved in Supplementary material but it is up to authors.

Author Response

ANSWERS TO REVIEWER 3

Reviewer #3: Comments to the Authors

Comment 3.1.: The introduction provides sufficient background (my only suggestion is add in this section some references to works related to synthetic potential antitumor drugs and their precursors: Journal of Molecular Structure 2016 1111, 142-150; New Journal of Chemistry 2017 41 (9), 3246-3250.

Authors: We thank very much the reviewer for the comment. The suggested papers [1,2] have been taken in consideration. However, in order to attend the reviewer proposal, a recently published review about the importance of natural products in the treatment of several diseases, mentioning the synthetic potential for new drugs development of natural compounds [3], was included in the text.

Comment 3.2.: May be, section 3.5.3. Isolation and chemical characterization procedure could be moved in Supplementary material but it is up to authors.

Authors: We thank the reviewer for the comment. We evaluated the proposal but, since the manuscript concerns a phytochemistry work which involves extraction, fractionation, isolation and compound characterization, we think that this paragraph would be more valorized in the methodology section of the manuscript than in the Supplementary Material.

Bibliography

  1. Kulish, K.I.; Novikov, A.S.; Tolstoy, P.M.; Bolotin, D.S.; Bokach, N.A.; Zolotarev, A.A.; Kukushkin, V.Yu. Solid State and Dynamic Solution Structures of O-Carbamidine Amidoximes Gives Further Insight into the Mechanism of Zinc(II)-Mediated Generation of 1,2,4-Oxadiazoles. Journal of Molecular Structure 2016, 1111, 142–150, doi:https://doi.org/10.1016/j.molstruc.2016.01.038.
  2. Anisimova, T.B.; Kinzhalov, M.A.; Guedes da Silva, M.F.C.; Novikov, A.S.; Kukushkin, V.Yu.; Pombeiro, A.J.L.; Luzyanin, K. v Addition of N-Nucleophiles to Gold(Iii)-Bound Isocyanides Leading to Short-Lived Gold(Iii) Acyclic Diaminocarbene Complexes. New Journal of Chemistry 2017, 41, 3246–3250, doi:10.1039/C7NJ00529F.
  3. Atanasov, A.G.; Zotchev, S.B.; Dirsch, V.M.; Orhan, I.E.; Banach, M.; Rollinger, J.M.; Barreca, D.; Weckwerth, W.; Bauer, R.; Bayer, E.A.; et al. Natural Products in Drug Discovery: Advances and Opportunities. Nature Reviews Drug Discovery 2021, 20, 200–216, doi:10.1038/s41573-020-00114-z.